# From Diagnosis to Survivorship: The Role of Social Determinants in Cancer Care

**DOI:** 10.3390/cancers17071067

**Published:** 2025-03-22

**Authors:** Abiha Abdullah, Zeyu Liu, Michele Molinari

**Affiliations:** 1Department of Surgery, University of Pittsburgh Medical Center, Pittsburgh, PA 15213, USA; 2Department of Surgery, University of California, Los Angeles, CA 90095, USA; zeyuliu@mednet.ucla.edu

**Keywords:** cancer disparities, social determinants of health, health equity, oncology access, cancer care interventions

## Abstract

Cancer care is shaped by social and economic factors that impact access and outcomes, particularly for underserved communities. This review examines disparities related to socioeconomic status, race, and geography, highlighting interventions to improve equity in cancer treatment and survivorship.

## 1. Introduction

SDOH—the conditions in which individuals are born, grow, live, work, and age—play a crucial role in shaping health outcomes across diverse populations [1,2,3]. Key SDOH factors—including socioeconomic status (SES), race, ethnicity, geographic location, housing and food security, education, and healthcare access—influence every stage of cancer care, from prevention and early detection to treatment and survivorship (Figure 1).

These disparities disproportionately affect vulnerable populations, leading to worse prognoses and survival rates [4,5].

Despite advancements in cancer treatment, improvements in survival remain unevenly distributed. More than 17 million cancer survivors in the U.S. contend with long-term treatment effects, such as chronic pain, emotional distress, and financial burdens, which are exacerbated by limited healthcare access [6,7]. Addressing these inequities requires policy-driven solutions, including Medicaid expansion, patient navigation programs, increased workforce diversity, and greater minority representation in clinical trials. Integrating SDOH screening into oncology care can help identify at-risk patients and connect them with critical resources such as transportation assistance, housing support, and financial counseling. Investments in telemedicine can further bridge gaps in care, particularly for rural and low-income populations.

This review examines the impact of SDOH on cancer care in North America, highlights key disparities, and presents evidence-based interventions to promote equitable treatment. A healthcare system that prioritizes SDOH-driven strategies can ensure that all patients—regardless of background—benefit from medical advancements, fostering a more just and inclusive future in oncology (Table 1).

## 2. Age, Gender, and Sexual Orientation

Cancer care disparities vary across demographic groups, affecting access, treatment, and outcomes. Pediatric and adolescent patients experience unique challenges due to developmental disruptions and the emotional and financial burdens on their families. The cost of treating adolescents in pediatric institutions is often higher than in adult settings due to increased resource utilization [8]. Limited access to specialized pediatric oncology care and psychosocial support further exacerbates disparities. Multidisciplinary approaches incorporating mental health providers and child life specialists have shown promise but these interventions remain inconsistently implemented, particularly in underserved areas [9].

Although survival has improved for several cancers in recent years, these improvements have been less pronounced for older individuals. Compounding factors such as functional decline, polypharmacy—associated with a 30% higher risk of treatment-related complications—and physiological age-related changes, including decreased organ function that increases the likelihood of undertreatment by 25%, further contribute to poorer outcomes. The lack of enrollment in clinical trials exacerbates these disparities, with only 25% of trial participants aged 65 or older, despite this group accounting for over 60% of cancer cases, limiting the generalizability of trial results [10,11]. Tools like geriatric assessment (GA), enhanced with SDOH evaluations, have demonstrated their effectiveness in reducing treatment-related toxicity by 20% and improving survival by 10% but their limited adoption, especially in underserved settings, perpetuates inequities. Widespread implementation of GA-guided care and multidisciplinary support is essential to address the unique needs of this vulnerable population [12,13,14,15,16,17].

Gender also plays a significant role in cancer care disparities. Societal expectations contribute to lower preventive care utilization among men, particularly for prostate and colorectal cancers. Conversely, women in survivorship report greater psychological distress and financial toxicity, often due to wage disparities and increased caregiving responsibilities. Gender differences also affect treatment outcomes; for example, men receive earlier diagnoses and guideline-concordant care more frequently for bladder cancer, while women are more likely to present with advanced disease and face poorer prognoses [18].

Lesbian, gay, bisexual, transgender, and queer (LGBTQ+) patients encounter significant barriers to equitable cancer care, including stigma, discrimination, and lack of culturally competent healthcare environments, which research has shown contribute to delayed diagnoses, compromised care quality, and worse outcomes. A study found that LGBTQ+ cancer survivors experience higher levels of discrimination, particularly among older patients, leading to medical mistrust and lower engagement in cancer care [19]. Another study demonstrated that nearly half (44.9%) of LGBTQ+ individuals reported experiencing discrimination in medical settings, which was associated with lower cancer screening rates and a higher likelihood of delaying care [20]. Lesbian and bisexual women often have lower breast and cervical cancer screening rates due to a lack of routine gynecological care and negative healthcare experiences, while transgender individuals face challenges like gender identity mismatches, which deter screenings for cancers such as prostate or breast cancer [21,22]. Systemic inequities in healthcare policies further exacerbate these disparities, with fragmented legislative frameworks failing to provide clear protections or guarantee access to comprehensive care. Expanding initiatives like the Healthcare Equality Index (HEI), which assesses institutional commitment to LGBTQ+ equity and nondiscrimination policies, could promote more inclusive healthcare environments. However, the direct impact of such programs on patient outcomes remains unclear [23]. Additionally, the lack of standardized gender identity data in cancer registries and clinical trials obscures disparities, impeding the development of tailored interventions [24]. Increased investments in LGBTQ+-specific research and public health initiatives are essential to addressing these gaps. Current funding remains disproportionately low, with only 0.1% of the National Institute of Health (NIH) funding in 2012 allocated to LGBTQ+ health outside HIV/AIDS [4,25]. Addressing these disparities requires comprehensive policy changes, enhanced provider training, and robust research efforts to ensure equitable cancer care for all populations.

## 3. Race and Ethnicity

Racial and ethnic disparities in cancer care stem from systemic inequities, structural barriers, and historical discriminatory practices. Studies in the U.S. show that Black patients are 67% less likely to receive guideline-concordant care for lung, colorectal, and breast cancers and experience significant delays in surgical treatment. They also face disproportionally higher mortality rates, with Black women being 40% more likely to die from breast cancer and Black men facing a 2.5-fold higher mortality rate for prostate cancer compared to their White counterparts, even after adjusting for insurance and socioeconomic factors. Similarly, Hispanic individuals, who have the lowest health insurance rates among racial and ethnic groups (19% uninsured compared to 7% of non-Hispanic White patients), frequently experience delays in cancer screening and treatment due to a lack of coverage [26,27,28,29]. Uninsured individuals are more likely to present with advanced-stage cancers and have poorer survival outcomes, a challenge often linked to employment in industries that do not provide health benefits.

Structural racism exacerbates these disparities, with historical policies such as redlining and unequal resource allocation segregating minority populations into areas with limited healthcare infrastructure, reducing access to timely and high-quality treatment [30,31]. Access to advanced cancer care is critical because emerging research highlights differences in inherited genomic variants among racial and ethnic groups [32,33]. For example, EGFR, BRAF, and ALK mutations are less common in Black patients compared to White, Asian, and Hispanic patients, while BRCA2 and PALB2 mutations in breast cancer appear more frequent in Black patients. Furthermore, breast cancers in Black women demonstrate higher genomic instability and increased expression of cancer-related pathways compared to those in White women. This underscores the need for greater diversity in genomic research and reference sequences to improve targeted therapies and patient outcomes [11].

Despite this growing evidence, systemic barriers persist, limiting equitable access to precision medicine and personalized cancer treatments. Additionally, racial and ethnic minorities remain underrepresented in clinical trials. A study evaluating 230 trials that led to U.S. Food and Drug Administration (FDA) approvals for oncology drugs between 2008 and 2018 found that White patients accounted for 76.3% of trial participants compared to 18.3% for Asian, 3.1% for Black, and 6.1% for Hispanic patients [34]. Institutions like the NIH have launched community-based outreach programs in partnership with local organizations to recruit minority populations, fostering trust and engagement in historically underrepresented communities [35]. The FDA and NIH also mandate demographic subgroup reporting in clinical trials to improve transparency and accountability, while initiatives like the Diversity in Clinical Trials Initiative encourage quota-based recruitment strategies to bridge racial and ethnic disparities in treatment access [36]. Furthermore, to address financial and logistical barriers hindering minority participation in trials, programs such as the Merck Expanded Access Program and the Bristol-Myers Squibb Foundation provide travel stipends, childcare assistance, and remote monitoring options [37]. Hospitals like the Cleveland Clinic and the Memorial Sloan Kettering Cancer Center offer financial navigation services to reduce out-of-pocket expenses [38,39]. State-level initiatives, including the California Cancer Assistance Program and the New York State Medicaid Cancer Treatment Program, further support uninsured and low-income patients with co-pay assistance and free screenings [40,41,42]. By implementing targeted recruitment, policy reforms, and financial aid, stakeholders can build a more inclusive healthcare system that ensures equitable access to clinical trials and life-saving treatments for all populations.

## 4. Geographic Disparities

Geographic location plays a critical role in shaping access to cancer care and influencing outcomes, with rural populations facing significant challenges compared to their urban counterparts. Rural patients undergo 25% fewer preventative screenings, are 1.5 times more likely to present with advanced cancer stages, and experience 10–20% higher mortality rates [8,43,44,45,46,47,48,49]. Geographic isolation compounds these disparities as nearly one in five rural Americans resides over 60 miles from a medical oncologist, while more than half must travel this distance to access gynecologic oncology care. These travel burdens lead to delayed diagnoses and treatments, negatively impacting health outcomes and imposing financial strain from long-distance travel [46]. Transportation barriers—including limited public transit, restricted access to private vehicles, and a shortage of healthcare facilities—further hinder timely follow-ups and treatment adherence. These logistical obstacles also contribute to disparities in clinical trial participation, with only 3% of rural patients enrolling in trials compared to 10% of urban patients, limiting their access to innovative therapies and research opportunities [45]. Implementing pragmatic solutions such as reducing travel requirements, expanding transportation support programs, and increasing the availability of clinical trials in community-based settings can help bridge these gaps and enhance access for rural patients. Telemedicine has emerged as a promising strategy to mitigate some of these geographic disparities by enabling remote consultations and reducing travel burdens. However, its full potential is hindered by unequal access to reliable internet connectivity, particularly in low-income and remote areas. Studies indicate that broadband penetration is substantially lower in the most rural counties, with only 59.9% of residents having sufficient internet speeds to support telemedicine visits compared to 96.0% in urban areas [50]. Digital literacy also remains a significant hurdle. Many elderly and low-income patients lack the necessary technological skills to navigate telehealth platforms, contributing to the underutilization of these services [51]. Despite these limitations, telemedicine has demonstrated success in improving care delivery, ensuring more timely access to specialized oncology services, and minimizing treatment delays [47,48]. Fully addressing rural disparities requires a comprehensive approach, including the expansion of telehealth services, the implementation of patient navigation programs, and policy reforms that prioritize investment in rural healthcare infrastructure, transportation networks, and digital health technologies [43].

## 5. Psychosocial and Behavioral Health Factors

Behavioral risk factors—including smoking, poor nutrition, physical inactivity, and excessive alcohol consumption—play a critical role in cancer incidence and outcomes, particularly among low-income populations. Smoking—responsible for 30% of all cancer-related deaths in the U.S.—is disproportionately prevalent in lower-income groups, with 25% of adults below the poverty line smoking compared to 14% of higher-income individuals [52,53]. This disparity contributes to increased rates of lung, throat, and other smoking-related cancers. Community-based interventions, such as the Charlotte Racial and Ethnic Approaches to Community Health (REACH) project, have effectively reduced smoking rates by 15% over five years through lay health advisors and tobacco policy reforms, including excise tax increases [53,54].

Poor nutrition and physical inactivity, both linked as associative factors to colorectal, breast, and endometrial cancers, are exacerbated by limited access to healthy foods and safe recreational spaces. Only 27% of adults below the poverty line meet physical activity guidelines compared to 45% of higher-income individuals [54]. Interventions such as community-based peer-led physical activity interventions have been shown to increase moderate-to-vigorous physical activity among low-income and minority older adults [55].

Similarly, excessive alcohol consumption—associated with liver, breast, and colorectal cancers—is more common in lower-income populations, with 8% of individuals in the lowest income bracket reporting heavy alcohol use compared to 5% in the highest [42]. Public health efforts, such as integrated alcohol reduction counseling programs, have reduced heavy drinking by 7% in underserved communities [53,54]. Effectively addressing these behavioral risk factors requires a comprehensive approach that integrates community engagement, policy reforms, and behavioral health counseling within oncology workflows. Strengthening these efforts will be essential to mitigating disparities and improving cancer prevention and outcomes in vulnerable populations.

## 6. Mental Health

Psychosocial factors, including stress, anxiety, and depression, profoundly influence cancer care outcomes [9]. The National Comprehensive Cancer Network (NCCN) identifies distress as a key risk factor for treatment nonadherence, which reduces both quality of life and survival rates [28]. Approximately 45% of cancer patients experience significant psychological distress, while 35% report depressive symptoms during active treatment—both of which are linked to longer treatment delays and worse clinical outcomes. A study at a safety-net hospital found that elevated distress correlated with treatment initiation delays exceeding 60 days, increasing disease progression and mortality risk [9,56].

Recognizing the impact of mental health on cancer care, the Society for Integrative Oncology (SIO) and the American Society of Clinical Oncology (ASCO) recommend evidence-based integrative therapies to address anxiety and depression throughout the cancer continuum. Interventions such as mindfulness-based stress reduction (MBSR), yoga, music therapy, tai chi/qigong, and reflexology have been shown to alleviate psychological distress [56]. For instance, MBSR programs have been shown to reduce anxiety by 25% and depressive symptoms by 15% in cancer patients. These therapies provide viable, patient-centered approaches, particularly for those facing barriers to traditional mental health care, however, their efficacy needs to be investigated further.

Furthermore, systemic inequities limit access to these interventions. Under-resourced patients are 30% less likely to receive timely treatment for anxiety and depression due to financial barriers, inadequate insurance coverage, and geographic disparities [9,57]. To address these gaps, embedding mental health screenings and integrative therapies into oncology care settings—as recommended by the SIO-ASCO guidelines—is critical. Additionally, policy interventions to expand insurance coverage and stigma reduction initiatives are essential to ensuring equitable access to psychosocial care, ultimately improving cancer treatment adherence and patient outcomes.

## 7. Economic Stability

Socioeconomic status (SES) significantly impacts cancer outcomes, with lower SES populations experiencing higher rates of advanced-stage cancer diagnoses and increased mortality. Patients from the lowest-income areas face a 13% higher risk of death compared to those from the highest-income areas. Additionally, individuals with less than 12 years of education have 1.6–2.8 times higher cancer mortality rates than those with 16 or more years of education [58,59]. Socioeconomic status (SES) significantly impacts cancer outcomes, with lower SES populations experiencing higher rates of advanced-stage cancer diagnoses and increased mortality. Patients from the lowest-income areas face a 13% higher risk of death compared to those from the highest-income areas. Additionally, individuals with less than 12 years of education have 1.6–2.8 times higher cancer mortality rates than those with 16 or more years of education [60].

Cancer screening rates are lower in low-income populations, exacerbating late-stage diagnoses. Only 27% of low-income adults adhere to colorectal cancer screening guidelines compared to 45% of higher-income individuals (48). Community outreach initiatives, such as the Cancer-Community Awareness Access Research and Education Project and the National Cancer Institute’s Screen to Save program, have increased screening rates by 10–15% in targeted communities [61,62,63].

Integrating SDOH data into electronic health records (EHRs) at institutions such as Boston Medical Center and South Carolina Health System has enhanced the identification of at-risk populations. Patients with multiple unfavorable SDOH face a 6.34 times higher risk of all-cause mortality. These efforts enable targeted interventions that improve patient outcomes and reduce disparities [64,65]. Addressing SES-related disparities in cancer care requires a comprehensive strategy that includes community outreach programs to improve screening and early detection, SDOH integration into EHRs to identify high-risk patients and tailor interventions, expanded insurance coverage to improve access to advanced therapies, and policy reforms to eliminate structural barriers in cancer care. By combining community engagement, data-driven interventions, and policy changes, healthcare systems can reduce SES-related disparities and improve cancer outcomes for vulnerable populations.

## 8. Neighborhood and Environment

Housing instability significantly disrupts cancer care by delaying diagnosis and treatment, leading to more advanced-stage presentations and poorer survival outcomes. Logistical barriers, such as a lack of transportation, difficulty communicating with providers, and prioritizing basic needs over medical care, often compromise treatment continuity. A study at a safety-net hospital found that patients facing housing instability and psychological distress experienced a median delay of 29 days from diagnosis to treatment initiation, compared to 22 days for those without distress [66]. Additionally, 44% of homeless individuals present with stage III or IV cancer, compared to 36% of housed patients. Among lung cancer patients, those experiencing housing instability have a 5-year survival rate of 15%, compared to 21% for those with stable housing, while colorectal cancer patients with housing instability face a 25% higher all-cause mortality rate [67,68,69]. Patients with unstable housing are also 1.5 times more likely to miss appointments and twice as likely to be nonadherent to treatment protocols.

Interventions such as housing resource referrals and patient navigator programs have shown promise in mitigating these disparities. Housing referrals have led to a 20% reduction in missed appointments and a 15% improvement in treatment adherence, while patient navigators have reduced treatment initiation delays by 10 days and improved overall survival rates by 5% among housing-insecure patients. Federal housing assistance programs have also been linked to earlier-stage cancer diagnoses [70,71,72,73]. However, housing instability often intersects with food insecurity, compounding barriers to cancer care and emphasizing the need for integrated, systemic solutions.

Malnourishment in cancer patients is a multifaceted issue affecting outcomes, influenced both by disease-related factors and SDOH. Disease-related malnourishment—commonly observed in cancer cachexia—arises from metabolic changes induced by tumor progression, inflammation, and treatment side effects, leading to significant weight loss and poor treatment. In contrast, food insecurity—a key social determinant—disproportionately affects low-income and minority populations, limiting access to nutritious foods essential for cancer prevention and treatment adherence. Residents in food deserts—areas with limited access to affordable, nutritious food—frequently rely on processed, unhealthy foods, contributing to higher obesity rates, which increase the risk of colorectal, endometrial, and breast cancers. Residents of food deserts are 32% more likely to be obese than those with access to nutritious food [74]. Furthermore, malnourished cancer patients are 2–3 times more likely to experience treatment-related complications, leading to reduced treatment efficacy and worse overall survival rates [75,76].

Policy-driven and community-based initiatives have improved food security and dietary quality in underserved areas. The Healthy Food Financing Initiative (HFFI) has increased access to nutritious foods, leading to a 10–15% rise in fruit and vegetable consumption among low-income populations. The expanding Supplemental Nutrition Assistance Program (SNAP) benefits have enhanced diet quality, with SNAP participants being 23% more likely to meet dietary guidelines for fruit and vegetable intake [77,78,79,80].

Cancer-focused interventions, such as the “Food to Overcome Outcomes Disparities” (FOOD) program and the ALCANCE Food for Health Equity program, have further addressed nutritional needs through food vouchers, grocery delivery, and culturally tailored initiatives. These programs have reduced treatment interruptions by 20%, improved nutritional status by 15%, increased treatment adherence by 25%, and enhanced quality of life by 30% among participants [81,82]. Despite these successes, broader and more sustainable strategies are needed to ensure equitable access to nutrition for all cancer patients.

Ultimately, integrating housing and food security into the broader framework of cancer care is essential. Embedding SDOH screening tools into healthcare workflows, strengthening community-based support programs, and advocating for federal policies that address the root causes of housing and food insecurity are critical steps in reducing disparities. Addressing these interrelated barriers will ensure more equitable access to cancer prevention and treatment, alongside improving long-term outcomes.

## 9. Environmental and Occupational Exposures

Low SES is strongly linked to occupational and environmental exposures that elevate cancer risk, disproportionately affecting marginalized populations. Workers in industries such as manufacturing, agriculture, and construction frequently encounter carcinogens, including asbestos, benzene, and pesticides [83].

In the National Lung Screening Trial, 28% of participants reported occupational exposures, with African Americans facing higher risks from silica and asbestos, contributing to increased odds of lung cancer diagnoses (adjusted odds ratio = 1.24–1.25) [84].

Environmental disparities further compound these risks. African American and Hispanic communities are disproportionately situated near industrial zones emitting known carcinogens such as benzene, 1,3-butadiene, and ethylene oxide. The likelihood of residing in areas with the highest benzene and ethylene oxide emissions is 10–20% higher for African Americans and 16–21% higher for Hispanics [84,85]. These exposures contribute to elevated incidences of lung, bladder, and mesothelioma cancers, with lung cancer rates notably higher among African Americans (4.3%) compared to White individuals (3.9%) [84].

Targeted interventions and policy reforms have significantly reduced cancer risks in these populations. The Clean Air Act led to a 73% reduction in emissions of common pollutants from 1970 to 2020, correlating with a 15–20% decline in lung cancer mortality in low-income and minority communities. Enhanced environmental health monitoring has helped identify pollution hotspots, resulting in a 10–15% reduction in cancer incidence in affected areas. Stricter workplace safety regulations enforced by the Occupational Safety and Health Administration (OSHA) have further decreased carcinogen exposure, leading to a 30% drop in mesothelioma cases and a 25% decline in bladder cancer incidence among high-risk workers [85,86,87,88]. Despite federal regulations, environmental pollution continues to disproportionately affect minority and low-income communities. Additionally, localized efforts, such as the California Air Resources Board (CARB) Community Air Protection Program, have provided real-time pollution monitoring and community engagement strategies to reduce cancer-related environmental risks [89]. While these measures demonstrate substantial progress, continued efforts are critical to addressing persistent disparities. Strengthening environmental regulations, enforcing occupational safety standards, and expanding access to preventive healthcare are essential to ensuring equitable cancer risk reduction across all populations.

## 10. Education and Health Literacy

Language barriers and low health literacy significantly hinder patient–provider communication, treatment adherence, and cancer outcomes, particularly among rural and undocumented immigrant populations with limited English proficiency (LEP) and lower socioeconomic status. LEP patients often struggle to understand medical information, leading to miscommunication, delays in diagnosis and treatment, and reduced adherence to care plans. For instance, Spanish-speaking caregivers of pediatric cancer patients report that language barriers prolong diagnostic and treatment timelines, with LEP patients being 20–30% more likely to experience treatment initiation delays [88,90,91]. Interventions that enhance communication improve outcomes. Employing trained interpreters or bilingual providers reduces communication errors by 42% and increases patient satisfaction by 26%, fostering better comprehension of diagnoses and care plans [91]. Additionally, culturally and linguistically aligned community health worker programs have boosted colorectal and breast cancer screening rates by 15–20% and 10–15%, respectively. Health literacy initiatives further enhance adherence to treatment protocols, improving overall outcomes in underserved populations [56,88].

Higher health literacy is strongly linked to better cancer care outcomes, including increased participation in screening programs. A meta-analysis found that individuals with adequate health literacy were significantly more likely to undergo breast, cervical, and colorectal cancer screenings, with adjusted odds ratios of 1.73, 1.64, and 1.25, respectively. Early detection through screening improves survival and treatment outcomes. Conversely, low health literacy is associated with poorer understanding of cancer-related information, reduced adherence to treatment plans, and lower quality of life. Patients with lower health literacy are 1.5 times more likely to miss appointments and twice as likely to be nonadherent to treatment protocols, further exacerbating health disparities [91].

Tailored health literacy programs have demonstrated measurable success. Interventions using plain language, visual aids, and culturally relevant materials have increased adherence to cancer treatment by 20% and improved overall quality of life by 15% [88,92,93]. These findings underscore the critical role of education and culturally tailored interventions in bridging communication gaps and improving cancer care outcomes, particularly for low-income and minority populations.

## 11. Financial Toxicity in Cancer Patients

Financial toxicity (FT), the economic burden associated with cancer care, is a significant challenge affecting nearly half of all cancer patients. The impact is particularly severe among low-income, uninsured, minority, and rural populations who face disproportionate financial hardship. FT manifests in several ways, including direct economic strain (e.g., medical debt or loss of income), psychological distress (e.g., anxiety and depression), and detrimental coping mechanisms (e.g., delaying or forgoing treatment). These factors collectively reduce quality of life, impair treatment adherence, and ultimately worsen survival outcomes. Patients experiencing FT are significantly more likely to forgo essential therapies such as chemotherapy, leading to suboptimal outcomes [94,95,96,97].

The financial burden of cancer treatment often leads to devastating consequences. Studies indicate that 42.4% of cancer patients exhaust their entire life savings within two years of diagnosis, with minority populations facing even greater risks of food insecurity, utility shut offs, and homelessness [9,94,98,99]. These disparities highlight the urgent need for interventions to mitigate FT. One successful initiative is the implementation of hospital-based financial navigation programs. At four hospitals, trained financial navigators helped secure USD 39 million in financial aid between 2012 and 2016, significantly reducing out-of-pocket costs for patients while also mitigating financial losses for healthcare institutions [100]. However, Casey J. Allen’s research emphasizes the importance of addressing both patient and provider perspectives on treatment costs as financial concerns heavily influence care decisions and adherence [101].

A comprehensive approach is essential to reducing FT. Patient education on healthcare financing is critical; for example, financial literacy courses for newly diagnosed cancer patients have been shown to improve understanding of treatment costs, employment implications, and available financial resources. Additionally, financial navigation programs—led by social workers or nurse navigators—have demonstrated success by providing individualized financial assessments, guiding patients to support services, and facilitating access to financial assistance. Participants in these programs report significantly lower FT and improved treatment adherence [96,102,103,104]. For example, a targeted assistance program supporting single mothers with metastatic breast cancer provided rent and mortgage support for six months, significantly improving their emotional well-being and preventing homelessness. However, financial distress often resumed after assistance ended, highlighting the need for sustained interventions [105].

At the systemic level, policy reforms play a vital role in addressing FT. Medicaid expansion has been instrumental in improving access to care by increasing insurance coverage, allowing more patients to afford essential treatments, receive earlier diagnoses, and achieve better health outcomes [106,107,108,109]. Addressing FT requires a dual approach: patient-centered interventions—such as financial literacy courses and navigation programs—must be combined with systemic policy reforms, including Medicaid expansion. These strategies collectively alleviate the economic burden of cancer treatment, improve adherence, enhance quality of life, and ultimately lead to better survival outcomes.

## 12. Workforce Diversity

A diverse oncology workforce has been associated with improved patient–provider communication and increased trust among racial and ethnic minority patients [110,111,112]. While some studies suggest that patients may have better adherence and satisfaction when treated by racially concordant providers, the direct impact on clinical cancer outcomes remains an area of ongoing research.

However, the underrepresentation of minority oncologists limits the ability to provide culturally competent care, particularly for non-English-speaking patients who face language barriers and misunderstandings that can impact treatment outcomes [113]. A tendency for bias has been suggested as a contributing factor to disparities in cancer care, with some research indicating that non-Black oncologists may have shorter or less-supportive interactions with Black patients, possibly contributing to poorer outcomes. Moreover, diverse oncology teams are more likely to design research that targets health disparities and engage minority patients in clinical trials, which is crucial for advancing equitable treatments [114]. Efforts to improve workforce diversity include targeted recruitment programs, mentorship opportunities, and financial support for underrepresented medical students. Examples include institutional initiatives, such as the Radiation Oncology Mentorship Initiative (ROMI)**,** that provide structured mentorship and research opportunities for medical students—particularly those from underrepresented backgrounds—to foster diversity within the field of oncology. In addition, there is the American Society of Clinical Oncology (ASCO) Diversity in Oncology Initiative, which offers career development awards **and** mentorship opportunities to encourage underrepresented trainees to pursue oncology [115,116].

## 13. An Intersectional View of Cancer Disparities

Cancer care disparities are rarely caused by a single factor; instead, they arise from the interaction of multiple social determinants such as socioeconomic status, race, ethnicity, geographic location, gender, and immigration status. Intersectionality acknowledges that these overlapping factors create unique and often more severe barriers to care.

Understanding these interconnected challenges helps clinicians and policymakers develop more effective solutions. Programs that combine financial assistance with culturally tailored patient navigation or include community-based health workers in oncology teams have been particularly effective in reducing treatment barriers and improving patient outcomes. 

Policy changes must also reflect an intersectional approach to create lasting improvements in healthcare access and quality. Expanding Medicaid to cover more at-risk populations, increasing funding for rural healthcare facilities, and ensuring diverse representation in clinical trials are essential steps toward addressing systemic inequities. Furthermore, incorporating intersectional research can highlight care gaps, allowing for more targeted and effective policy interventions.

## 14. Conclusions

Achieving equity in cancer care demands a shift beyond the clinical setting to dismantle systemic barriers, expand access, and embrace innovative solutions. The strategies outlined—spanning policy reforms, community-based interventions, and technological advancements—offer a clear and actionable roadmap for transforming oncology care. Future efforts must integrate these approaches into standard practice, ensuring sustainable, patient-centered solutions that eliminate barriers and drive improved outcomes for all populations. By prioritizing inclusivity and innovation, the healthcare system can bridge existing gaps and move toward a future where high-quality cancer care is accessible to everyone, regardless of geographic location or socioeconomic status.

## Figures and Tables

**Figure 1 cancers-17-01067-f001:**
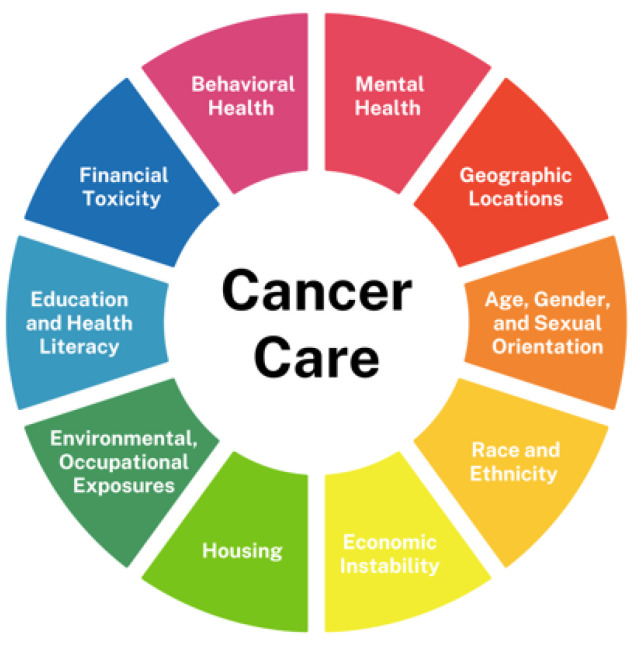
Social determinants of health in cancer care.

**Table 1 cancers-17-01067-t001:** Summary of key SDOH factors influencing cancer outcomes, illustrating disparities in diagnosis, treatment, and survival while outlining interventions such as policy reforms, community programs, and SDOH data integration to advance equitable cancer care.

SDOH	Key Impacts on Cancer Care	Key Examples	Interventions and Future Recommendations
**Socioeconomic status (SES)**	Lower SES increases advanced-stage diagnoses, cancer mortality, and barriers to treatment.	Patients in the lowest-income bracket have a **13% higher risk of death**. Low-income colorectal cancer patients are **40–60% less likely** to receive targeted therapies.	**Expand Medicaid coverage** and **reduce out-of-pocket costs** to improve access. **Community outreach initiatives** (e.g., **Screen to Save**) to increase screenings. **Integrate SDOH data into EHRs** to enable targeted interventions.
**Race and ethnicity**	Racial minorities experience delayed diagnoses, reduced access to quality care, and implicit bias in treatment decisions.	Black patients have a **21% higher mortality rate** from breast cancer. Hispanic patients are **30% less likely** to receive timely colorectal cancer screening.	**Mandate implicit bias training** for healthcare providers. **Increase clinical trial diversity** through targeted recruitment. **Expand Medicaid coverage** and **enhance culturally competent care**.
**Age**	Pediatric, adolescent, and elderly patients experience unique barriers in care access, treatment plans, and survivorship.	**Pediatric patients** face psychological distress and treatment toxicity. **Adolescents and Young Adults (AYAs)** are **underrepresented in clinical trials**. **Elderly patients** are **less likely to receive aggressive treatments** due to age-related biases.	**Provide age-specific survivorship care plans** and **peer support groups** for AYAs. **Mandate geriatric assessments (GAs)** to personalize cancer treatment.
**Gender and sexual orientation**	LGBTQ+ patients face stigma, discrimination, and culturally incompetent care, reducing screening rates and access to cancer treatment.	**Gay men** have higher rates of **HPV-related anal cancer**. **Lesbian/bisexual women** are **40% less likely** to receive mammograms. **Transgender patients** face **barriers to gender-affirming cancer care**.	**Develop LGBTQ+-specific provider training** to improve cancer care competency. **Ensure gender-affirming care coverage** in oncology settings. **Increase representation in cancer clinical trials** for LGBTQ+ patients.
**Geographic location**	Rural populations experience **higher cancer mortality**, reduced access to screenings, and longer travel distances to specialists.	Rural patients are **15–30% more likely** to present with late-stage cancer. Long travel distances contribute to **delayed chemotherapy and radiation therapy**.	**Expand telemedicine oncology services** to reduce travel burdens. **Subsidize rural specialty providers** and **increase funding for mobile cancer clinics**. **Improve transportation infrastructure** for rural patients.

## Data Availability

No datasets were generated or analyzed for this study.

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
