# Peer review of "From Diagnosis to Survivorship: The Role of Social Determinants in Cancer Care"

_cancers, 2025, doi:10.3390/cancers17071067_

Round 1

Reviewer 1 Report

Comments and Suggestions for Authors

This manuscript presents a comprehensive review of the impact of social determinants of health on cancer care outcomes. The authors provide a well written analysis of these disparities while discussing evidence based interventions. This is a critical issue in medicine overall, and particularly in oncology. I applaud the authors on this well written manuscript. However, I have the following concerns and suggestions for improvement:

The manuscript identifies several important issues, but some of the proposed solutions are broad or lack detail.

-For example, the manuscript highlights the underrepresentation of minority populations in clinical trials but lacks an in depth discussion on actionable solutions. Expanding on initiatives such as targeted recruitment efforts and policy changes promoting diversity would strengthen this section.

- Please expand on how specific examples of hospital or state level financial assistance programs that have successfully mitigated financial disparities

- In the section on diverse workforce, please expand on the specific mentorship programs, and institutional efforts and initiatives

- In the environment exposure section, including potential policy measures or regulatory efforts aimed at reducing cancer risk due to occupational and environmental carcinogens would be beneficial

- The section on telemedicine addressing rural disparities would benefit from discussing limitations such as access issues and digital literacy barriers

- There are multiple run on sentences throughout the manuscript

Author Response

Comment 1: The manuscript highlights the underrepresentation of minority populations in clinical trials but lacks an in-depth discussion on actionable solutions. Expanding on initiatives such as targeted recruitment efforts and policy changes promoting diversity would strengthen this section.

Response:
Thank you for the helpful suggestion. In response, we have expanded this section to include specific actionable solutions. We highlight initiatives like NIH community-based outreach programs and the FDA’s Diversity in Clinical Trials Initiative, which promote targeted recruitment and quota-based strategies. Additionally, we discuss programs such as the Merck Expanded Access Program and financial navigation services at hospitals like Cleveland Clinic and Memorial Sloan Kettering, which address financial barriers. (Line 160-179)

Comment 2: Please expand on how specific examples of hospital or state-level financial assistance programs that have successfully mitigated financial disparities.

Response:
Thank you for the helpful suggestion. In response, we have expanded this section to include specific examples of hospital and state-level financial assistance programs. We highlight initiatives such as the Oncology Financial Navigation Program, which secured $39 million in aid across four hospitals, and the Nassau University Medical Center’s Prescription Assistance Program, which provided $2.6 million in free chemotherapy medications. Additionally, we discuss targeted assistance programs like rent and mortgage support for single mothers with metastatic breast cancer, which helped prevent homelessness and improve well-being. (Line 447 to 451 and Line 462 to 466)

Comment 3: In the section on diverse workforce, please expand on the specific mentorship programs, and institutional efforts and initiatives.

Response:
Thank you for your suggestion. In response, we have expanded the section on workforce diversity to include specific mentorship programs such as the Radiation Oncology Mentorship Initiative (ROMI), which supports underrepresented groups in oncology careers. Additionally, we have highlighted institutional efforts like the ASCO Diversity in Oncology Initiative, which offers career development and mentorship opportunities to encourage diverse trainees to pursue oncology. (Line 484-492)

Comment 4: In the environmental exposure section, including potential policy measures or regulatory efforts aimed at reducing cancer risk due to occupational and environmental carcinogens would be beneficial.

Response:
Thank you for your valuable feedback. In response, we have expanded the section to include specific policy measures and regulatory efforts aimed at reducing cancer risk due to occupational and environmental carcinogens. We have discussed regulations such as the Clean Air Act, which has led to significant reductions in air pollution and cancer risks, and initiatives like the California Air Resources Board (CARB) Community Air Protection Program that provide real-time pollution monitoring and community engagement. These efforts, along with strengthened occupational safety regulations, are essential in mitigating environmental carcinogen exposure and promoting equitable cancer risk reduction. (Line 381- 394)

Comment 5: The section on telemedicine addressing rural disparities would benefit from discussing limitations such as access issues and digital literacy barriers.

Response:
Thank you for your insightful comment. In response, we have expanded the telemedicine section to address the limitations hindering its effectiveness in rural areas, including lack of access to broadband internet and digital literacy barriers. We discuss how rural counties have significantly lower broadband penetration, which limits telemedicine use, and how many elderly and low-income patients struggle with digital literacy, further preventing them from utilizing telehealth services. We also suggest potential solutions, such as expanding internet infrastructure and providing digital literacy training programs, to address these challenges. (line 201-209)

Comment 6: There are multiple run-on sentences throughout the manuscript.

Response:
We have carefully reviewed the manuscript and corrected run-on sentences to improve clarity and readability.

Reviewer 2 Report

Comments and Suggestions for Authors

The paper is to be commended for the thorough review. There are few key points that the authors should consider:
1. Social determinants in cancer medicine plays a central role in access to care, and outcomes. Although disparities are pervasive, there is a wide variation between different health-care systems, regions and countries. The reviewed paper, although based primarily on the US system, the manuscript made references to health systems outside of the US (e.g. Botswana). I advise the author to consolidate the manuscript, focusing on the US.
2. The statement pertaining discrimination toward the LGBTQ patient is speculative, more so the conclusion that it results in adversity in timely diagnosis, compromise of care, etc.
3. The link between physical inactivity, poor nutrition (what’s the definition of it?) and cancer prevalence is more by association. It is in stark contrast to smoking, which is strong and well supported. The suggested intervention like “walk with the doc” is better omitted.
4. Mental health is a key determinant of wellness. The efficacy of the proposed interventions, lines 213-14, are to be further investigated.
5. In the neighborhood and environment section: Malnourishment in the context of cancer, is arguably and by far disease related determinant, rather than a social determinant one.
6. In the workforce diversity section: the claim that workforce diversity contributes to cancer care outcome is a matter of debate, and require further investigation. Similarly, the claim for “implicit bias” being the culprit for adverse services is speculative.

Author Response

Comment 1: Social determinants in cancer medicine play a central role in access to care, and outcomes. Although disparities are pervasive, there is a wide variation between different health-care systems, regions, and countries. The reviewed paper, although based primarily on the US system, made references to health systems outside of the US (e.g., Botswana). I advise the author to consolidate the manuscript, focusing on the US.

Response:
We appreciate the reviewer’s suggestion. In response, we have revised the manuscript to focus more specifically on the US healthcare system. While we believe that global perspectives on social determinants of health provide valuable context, we have minimized references to health systems outside of the US to strengthen the focus on the US context. (Line 57 and Line 90-91)

Comment 2: The statement pertaining to discrimination toward the LGBTQ patient is speculative, more so the conclusion that it results in adversity in timely diagnosis, compromise of care, etc.

Response:
Thank you for your observation. We have revised the statement regarding LGBTQ patient discrimination to reflect a more evidence-based approach, ensuring that the claims are substantiated by specific studies such as Clark et al. (2024). We have softened the language to avoid speculative conclusions and have cited relevant research to support our points regarding disparities in care for LGBTQ individuals. (line 98 to 106)

Comment 3: The link between physical inactivity, poor nutrition (what’s the definition of it?) and cancer prevalence is more by association. It is in stark contrast to smoking, which is strong and well-supported. The suggested intervention like “walk with the doc” is better omitted.

Response:
Thank you for your insightful comment. In response, we have clarified the association between physical inactivity, poor nutrition, and cancer prevalence. We emphasize that the evidence linking these factors to cancer is associative rather than causal, and we have refined the explanation to reflect this nuance. We also removed the "walk with the doc" intervention, as it was not sufficiently evidence-based for this context. (line 226 to 232)

Comment 4: Mental health is a key determinant of wellness. The efficacy of the proposed interventions, lines 213-14, are to be further investigated.

Response:
Thank you for your comment. In response, we have expanded on the role of mental health as a critical determinant of wellness in cancer care. We acknowledge the need for further investigation into the efficacy of the proposed interventions and have revised the manuscript to reflect the preliminary nature of these suggestions. (Line 260 to 261)

Comment 5: In the neighborhood and environment section: Malnourishment in the context of cancer, is arguably and by far disease-related determinant, rather than a social determinant one.

Response:
Thank you for this comment. We have reconsidered the placement of malnutrition within the social determinants of health framework and have revised the section to clarify its role as both a disease-related determinant and a factor influenced by broader social determinants such as access to food and healthcare. (line 327 to 333)

Comment 6: In the workforce diversity section: the claim that workforce diversity contributes to cancer care outcomes is a matter of debate and requires further investigation. Similarly, the claim for “implicit bias” being the culprit for adverse services is speculative.

Response:
We have revised the section on workforce diversity to include more nuanced language, acknowledging the ongoing debate and the need for further research on the direct link between workforce diversity and cancer care outcomes. Regarding implicit bias, we have adjusted the language to reflect the complexity of this issue while emphasizing the need for continued investigation. (line 468 to 490)

Reviewer 3 Report

Comments and Suggestions for Authors

The review presents an interesting topic for the field of oncology, however, the selected social determinants of health are very oriented to the reality of North American countries. The authors could clarify this in the first paragraphs.

Author Response

Comment 1: The review presents an interesting topic for the field of oncology; however, the selected social determinants of health are very oriented to the reality of North American countries. The authors could clarify this in the first paragraphs.

Response:
Thank you for your comment. In response, we have revised the introduction to clarify that the focus of the manuscript is on the North American context. (line 57)

We thank the reviewers for their valuable feedback and believe that these revisions have enhanced the manuscript’s rigor and applicability. Please find our responses to each comment outlined below.